# Reconstructing the phylogeny and evolutionary history of freshwater fishes (Nemacheilidae) across Eurasia since early Eocene

**Vendula Bohlen Šlechtová[1], Tomáš Dvořák[1,2], Jörg Freyhof[3], Maurice Kottelat[4†], Boris Levin[5,6], Alexander Golubtsov[6], Vlastimil Šlechta[1], Joerg Bohlen[1]\***

[1]Laboratory of Fish Genetics, Institute of Animal Physiology and Genetics, Academy of Sciences of the Czech Republic, Libechov, Czech Republic; [2]Department of Zoology, Faculty of Science, Charles University, Prague, Czech Republic; [3]Museum für Naturkunde, Leibniz Institute for Evolution and Biodiversity Science, Berlin, Germany; [4]Lee Kong Chian Natural History Museum, National University of Singapore, Singapore, Singapore; [5]Papanin Institute of Biology of Inland Waters, Russian Academy of Sciences, Borok, Russian Federation; [6]A.N. Severtsov Institute of Ecology and Evolution of the Russian Academy of Sciences, Moscow, Russian Federation

**\*For correspondence:**
joerg_bohlen@yahoo.de

**Present address:** [†]Rue des Rauraques 6 (permanent address), Delémont, Switzerland

**Competing interest:** The authors declare that no competing interests exist.

## eLife Assessment

In this **important** study, the authors reconstruct the evolutionary history of a large and widespread group of freshwater fishes (Nemacheilidae) across Eurasia since the early Eocene, based on molecular phylogenetic analysis with very comprehensive samplings including 471 specimens belonging to 250 living species. The authors **convincingly** infer that range expansions of the family were facilitated by tectonic connections, favourable climatic conditions, and orogenic processes, adding to our understanding of the effects of climatic change on biodiversity during the Cenozoic. This work is of interest to evolutionary biologists, ichthyologists, paleontologists, and general readers.

**Abstract** Eurasia has undergone substantial tectonic, geological, and climatic changes throughout the Cenozoic, primarily associated with tectonic plate collisions and a global cooling trend. The evolution of present-day biodiversity unfolded in this dynamic environment, characterised by intricate interactions of abiotic factors. However, comprehensive, large-scale reconstructions illustrating the extent of these influences are lacking. We reconstructed the evolutionary history of the freshwater fish family Nemacheilidae across Eurasia and spanning most of the Cenozoic on the base of 471 specimens representing 279 species and 37 genera plus outgroup samples. Molecular phylogeny using six genes uncovered six major clades within the family, along with numerous unresolved taxonomic issues. Dating of cladogenetic events and ancestral range estimation traced the origin of Nemacheilidae to Indochina around 48 mya. Subsequently, one branch of Nemacheilidae colonised eastern, central, and northern Asia, as well as Europe, while another branch expanded into the Burmese region, the Indian subcontinent, the Near East, and northeast Africa. These expansions were facilitated by tectonic connections, favourable climatic conditions, and orogenic processes. Conversely, aridification emerged as the primary cause of extinction events. Our study marks the first comprehensive reconstruction of the evolution of Eurasian freshwater biodiversity on a continental scale and across deep geological time.

**eLife digest** Stone loaches, also known as Nemacheilidae, are a large family of fish commonly found in the streams and rivers of Europe and Asia, with a small number of species also inhabiting certain Ethiopian lakes.

How these fish, which originated in Asia about 50 million years ago, made their way into European and African waters remains poorly understood. Major geological and climate changes took place throughout this period, from the formation of the Himalayas and other mountain ranges in eastern and western Asia to large drops in temperatures or rainfalls in certain regions.

Šlechtová et al. studied the influence of these events on the spread and evolution of stone loaches. The team used a large dataset of 471 samples obtained from more than 250 species to reconstruct the evolutionary tree of the Nemacheilidae. The analysis uncovers six major groups (or clades) within the family, all stemming from a common ancestor living 48 million years ago in Indochina (current mainland Southeast Asia). Each clade has separate yet sometimes overlapping geographical distributions. They followed distinct routes to spread across Asia and Europe, which Šlechtová et al. were able to examine in the light of geological and climate changes.

For instance, a major aridification event taking place in Central Asia between 34 to 23 million years ago created a geographical divide within an ancestral stone loach group, splitting it into two parts that evolved separately to form two of the six current clades. While the Himalayas also acted as a strong barrier, growing highlands in eastern and western Asia expanded the range of suitable habitats for the fish, allowing them to colonize central and northern Asia and, from there, Europe.

Other major geological events played a strong role in the propagation of the Nemacheilidae. When a small tectonic plate known as West Burma Terrane first contacted Southeast Asia 33 million years ago and later northeast India around 30 million years ago, the ancestral fish family used the plate like a ferry boat to spread to these new territories, and from there, expand into the Near East, Southeast Europe and Northeast Africa.

These findings build on prior work investigating how geological and climate events have shaped evolution. However, they are the first case study to show the complete evolution of an animal group over such a large area and long period. It is the first detailed example of its type and could be precious to inform future work on evolution.

## Introduction

The present-day species- and genus-level diversity on earth is mainly the result of the evolution during Cenozoic, when biodiversity recovered from the Cretaceous-Tertiary mass extinction event, which eradicated three-quarters of all animal and plant species. During the Cenozoic new lifeforms flourished; often from lineages that had played only a minor role before the mass extinction, like mammals, birds, and teleost fishes (*Feduccia, 1995*; *Friedman, 2010*).

The Cenozoic is also a period of significant and relatively rapid climatic and geological changes (*Summerhayes, 2015*). The biggest geological changes occurred in Eurasia, resulting from its collision with the African, Arabian, and Indian plates, which caused the closure of the Tethys Sea and the Alpine–Himalayan orogeny (*Trifonov et al., 2012*). The climate also changed dramatically during Cenozoic; transitioning Earth from a warm and humid hothouse state to a cool and dry icehouse (*Westerhold et al., 2020*), but with a complex interplay of large-scale changes and smaller-scale fluctuations, creating a diverse range of climatic conditions on the local scale (*Mudelsee et al., 2014*).

Geological and climatic changes are major drivers of biodiversity evolution (*Antonelli et al., 2018*; *Fritz et al., 2016*; *Mayhew et al., 2012*). Numerous biogeographical studies show distribution patterns and local vicariance and radiation events in the Eurasian biodiversity (e.g. *Andreyenkova et al., 2021*; *Ishii et al., 2023*). The majority of these studies focused on terrestrial fauna and flora, but we believe that the best-suited model organisms for biogeographical analysis are strict freshwater species. Such hololimnic species are confined to a given water body unless a physical connection with another water body appears. At the same time, the evolution of the earth's hydrographic network is caused by local geomophological and climatic conditions. The locally 'locked' freshwater fauna thus reflects the environmental past of any region on earth in much more detail than most terrestrial animals or plants (e.g. *Bănărescu, 1992*; *Bohlen et al., 2020a*).

Despite this potential, there is a paucity of research on the evolutionary history of Eurasian freshwater biodiversity covering the large geographic and temporal scales. While numerous studies have examined certain geological or climatic changes during specific periods of the Cenozoic (e.g. *Ballarin and Li, 2018*; *Gu et al., 2022*; *Kahlke, 2014*) or within limited regions of Eurasia (e.g. *Delić et al., 2020*; *Deng et al., 2020*; *Saito et al., 2018*), a comprehensive analysis encompassing the whole Eurasia and through most of the Cenozoic era and considering climatic as well as geological events as driving forces of biodiversity evolution is lacking.

For the present study, we chose the species-rich (presently 838 species) freshwater fish family Nemacheilidae (stone loaches) as model, which is present in almost all water bodies of Europe and Asia (*Dvořák et al., 2022*; *Kottelat, 2012*) and stretches from Spain to Japan in west–east direction and from Russia to Indonesia in north–south direction. Most species are small (<12 cm SL), benthic and inhabit streams and rivers, although a few are found in swamps, caves, and lakes. Few even occupy extreme environments: Nemacheilidae include the deepest freshwater cave fish in the world (*Triplophysa gejiuensis*, 400 m below surface) and the fish living at the highest altitude (*Triplophysa stolickai*, 5200 m a.s.l.) (*Chu and Chen, 1979*; *Kottelat, 2012*).

The aim of this study is to reconstruct the evolutionary history of the family Nemacheilidae as example for the evolution of freshwater fauna across Eurasia and since early Cenozoic. The research concept includes a phylogenetic reconstruction based on molecular genetic data, followed by the dating of cladogenetic events and a biogeographic analysis in order to reconstruct ancestral distribution ranges. The results are compared with the known geological and climatic history of Eurasia to identify the causative factors and processes that have influenced the diversification of Eurasian freshwater fishes.

## Results

### Major clades within Nemacheilidae

To understand the phylogenetic diversity of the family Nemacheilidae and identify the largest units within, we analysed sequence data for one mitochondrial (cytochrome *b*) and up to five nuclear (EGR3, IRBP2, Myosin6, RAG1, Rhodopsin1) genes of 471 nemacheilid specimens representing 279 species and 37 genera from across Eurasia (*Figure 1*, *Supplementary file 1*). This is by far the most comprehensive genetic dataset of Nemacheilidae ever published. The outgroup included additional 21 species from nine related families (*Supplementary file 1*). We applied maximum likelihood (ML) analysis in IQ-TREE and Bayesian inference (BI) in MrBayes on the concatenated dataset. Besides, the individual unrooted ML gene trees were reconstructed in IQ-TREE and subsequently used as input files for ASTRAL III (*Zhang et al., 2018*) to infer species tree. All analyses revealed Nemacheilidae as monophyletic group and recovered the presence of six monophyletic major clades within the dataset (*Figure 2*).

Each major clade exhibits a distinct distribution *Figure 2*. The most widely distributed is the **'Northern Clade'** (*Figure 2A*), which inhabits the northern parts of Asia and most of Europe from Japan to Spain and southwards to Yunnan. In our analyses it includes all species of the genera *Barbatula*, *Claea*, and *Triplophysa*. The **'Southern Clade'** (*Figure 2A*) has the second largest distribution and inhabits the Indian subcontinent, the Burmese region and West Asia from the Salween River to southeast Europe; isolated species are found also in Indochina and East Africa (Ethiopia). The Southern Clade is the clade with the highest taxonomic diversity including all analysed species of 14 genera (*Afronemacheilus*, *Mesonoemacheilus*, *Mustura*, *Nemachilichthys*, *Neonoemacheilus*, *Oxynoemacheilus*, *Paracobitis*, *Paraschistura*, *Petruichthys*, *Physoschistura*, *Pteronemacheilus*, *Sasanidus*, *Seminoemacheilus*, and *Turcinoemacheilus*) plus several species of the genus *Schistura*. Another clade is the **'Burmese Clade'** (*Figure 2B*) with a continuous distribution from western Thailand to Northeast India, plus isolated species in the Western Ghats of India and in eastern Thailand and Cambodia. It comprises all analysed species of the genera *Aborichthys*, *Acanthocobitis*, and *Paracanthocobitis*, as well as several species classified as *Schistura*. The **'Sundaic Clade'** (*Figure 2B*) includes only the species of the genus *Nemacheilus* and is distributed across Sundaland (the Great Sunda Islands Borneo, Sumatra and Java, and the Malay Peninsula) and in the Indochinese rivers draining to the Gulf of Thailand (Mekong, Mae Khlong, and Chao Phraya). The **'Eastern Clade'** (*Figure 2C*) is found along the eastern margin of Asia from the Russian Far East to eastern Borneo. Its disjunct distribution

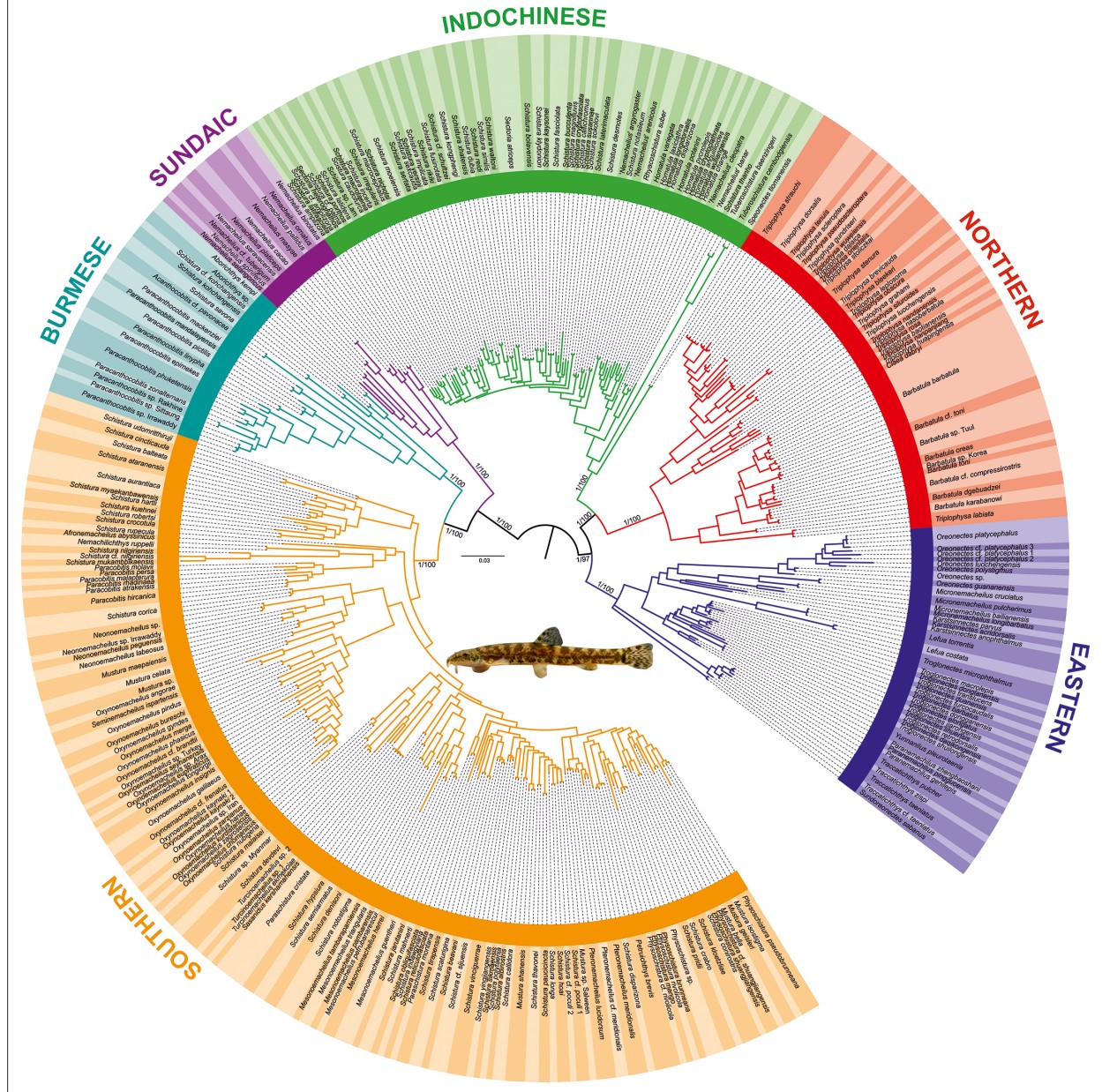

**Figure 1.** Phylogenetic tree of Nemacheilidae, including all 471 analysed nemacheilid specimens, reconstructed in MrBayes based on six loci. The topologies of Bayesian inference (BI) and maximum likelihood (ML) trees were congruent. The values at the basal nodes correspond to Bayesian posterior probabilities and ML bootstrap supports, respectively. Only the ingroup is shown. The nemacheilid pictured in the centre is *Barbatula* sp. Korea.

consists of a single genus (*Sundoreonectes*) in northeast Borneo, the majority of genera and species in northern Vietnam and southeast China and a single genus (*Lefua*) on the Korean Peninsula, the Japanese Archipelago, northeast China, and the Russian Far East. The Eastern Clade comprises all analysed species of the genera *Karstsinnectes*, *Lefua*, *Micronemacheilus*, *Oreonectes*, *Paranemachilus*, *Sundoreonectes*, *Traccatichthys*, *Troglonectes*, and *Yunnanilus*. The sixth clade ('**Indochinese Clade**', *Figure 2C*) has a continuous distribution throughout Thailand, Laos, Cambodia, Vietnam, central Myanmar and southern China, as well as isolated species in the upper/middle Yangtze and on Tioman Island (Malaysia). It is composed of all analysed species belonging to *Homatula*, *Rhyacoschistura*, *Sectoria*, *Spaeonectes*, and *Tuberoschistura*, in addition to many species currently assigned to *Schistura* and four species of uncertain generic assignment ('*Nemacheilus*').

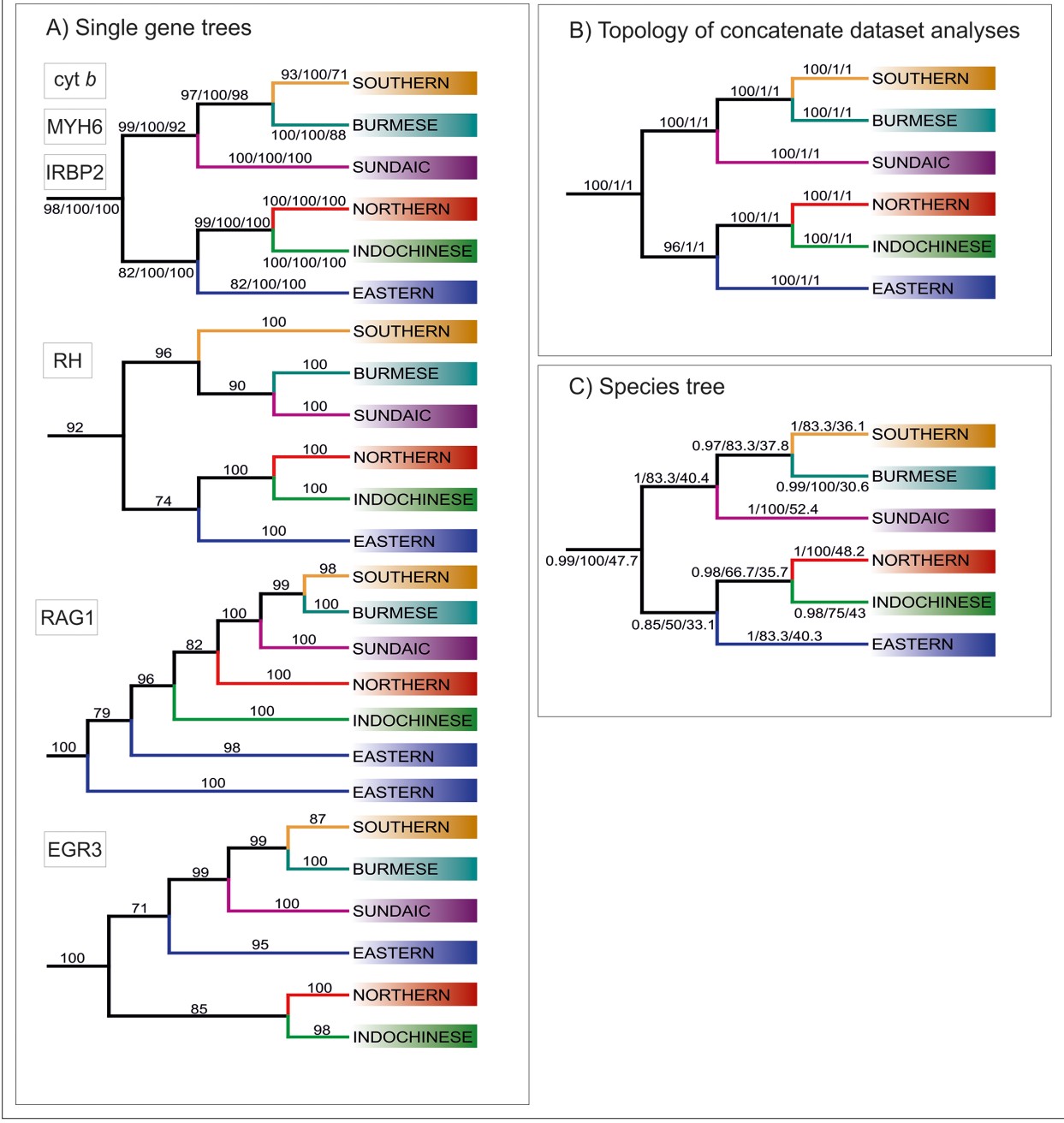

**Figure 2.** Topologies of main clades derived from various analyses. (**A**) Topologies recovered by ML analyses of single gene trees, with values at branches indicating bootstrap supports. Topologies congruent to the concatenated data-derived as well as species tree is revealed by Cyt b, MYH6 and IRBP2. Three slightly different topologies were derived from RH, EGR3 and RAG1 (**B**) Topology obtained from analyses of concatenate dataset, the values at the branches correspond to ML/MrBayes PP/BEAST PP supports, respectively. (**C**) Topology inferred by ASTRAL-III using the unrooted ML single gene trees as input. The values at the branches correspond to support values/gCF/sCF.

The distribution areas of the six major clades are not mutually exclusive. While the Northern, Indochinese, and Southern Clades exhibit limited overlap with each other, the Sundaic Clade shares its northern range with the Indochinese Clade. The Eastern Clade overlaps with the Northern Clade in the north, the Indochinese Clade in the middle, and the Sundaic Clade in the south. The Burmes Clade shares most of its distribution area with the Southern Clade.

Analyses of the concatenated dataset as well as ASTRAL analysis inferring a species tree from single gene trees revealed a basal dichotomy in the family Nemacheilidae (*Figure 1*; *Figure 2* and

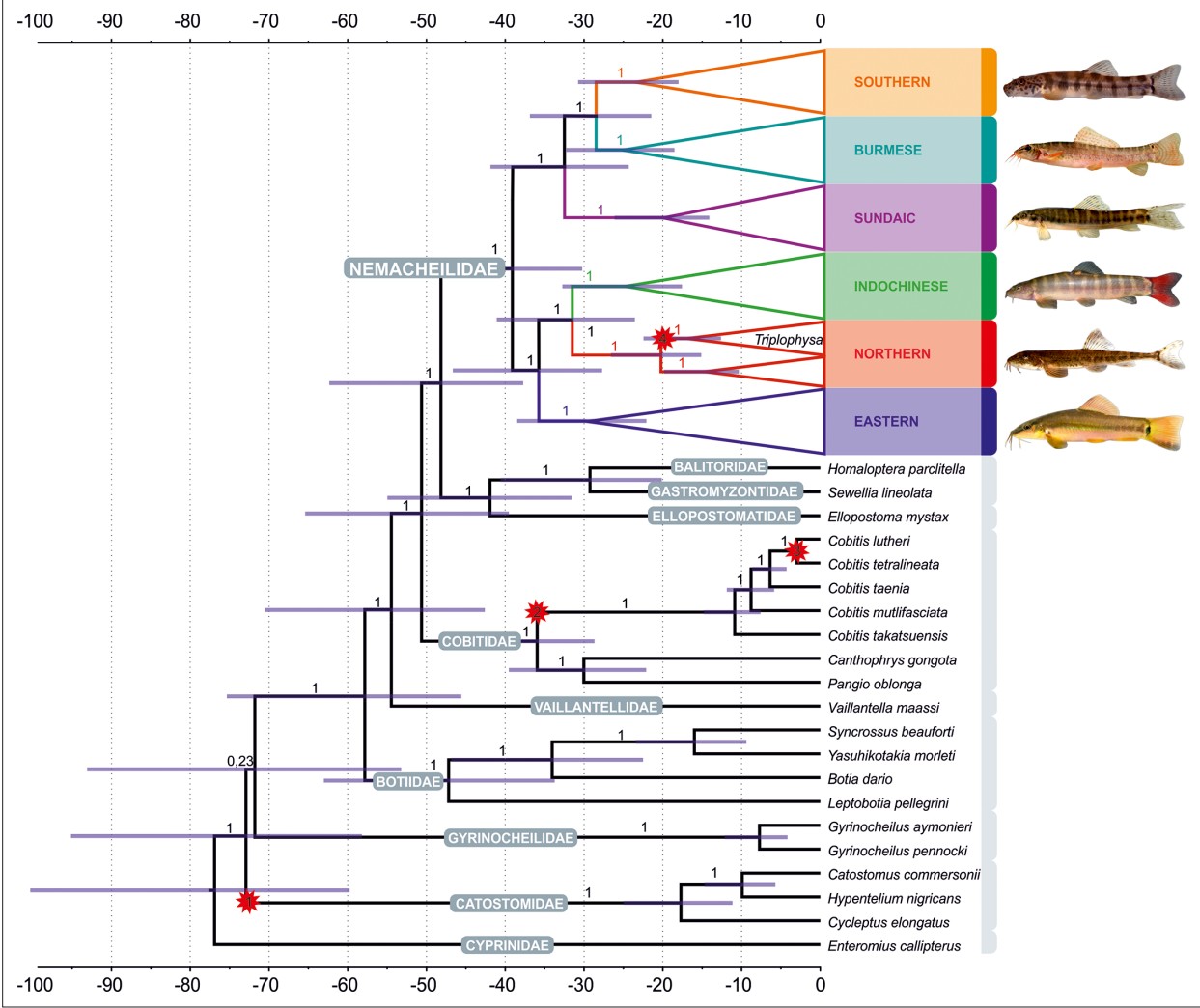

**Figure 3.** Divergence time estimation resulting from Bayesian divergence time analysis of concatenated dataset in BEAST 2: maximum clade credibility (MCC) tree of the whole dataset, with the Nemacheilidae ingroup collapsed into main clades. Red stars indicate calibration points, values at branches represent posterior probabilities (PPs lower than 0.90 not shown) and blue horizontal bars denote 95% HPD. The time scale is in millions of years. Pictures on the right show one example species for each clade; from top: *Schistura hypsiura* (Southern Clade), *Paracanthocobitis pictilis* (Burmese Clade), *Nemacheilus chrysolaimos* (Sundaic Clade), *Schistura yersini* (Indochinese Clade), *Triplophysa stenura* (Northern Clade), and *Traccatichthys taeniatus* (Eastern Clade).

*Figure 3*). The one branch consists of the Eastern Clade as sister to the pair Indochinese plus Northern Clade. The second branch is made of the Sundaic Clade as sister to the pair Burmese and Southern Clade. All basal nodes and most of tip nodes are supported by high statistical supports in all analyses.

## Divergence times

The ages of the genealogical events were determined using three fossil records and one geological event as calibration points (details of calibration under 'Methods'). The results indicate that Nemacheilidae shared the last common ancestor with other families of Cobitoidea about 48.5 mya (*Figure 4*). The first notable split that separated the family into two big branches took place around 39 mya (for details, see *Figure 5*).

## Biogeographic reconstructions

The biogeographic reconstruction of Nemacheilidae considered 11 geographic regions within Eurasia and northeast Africa. The results show that Nemacheilidae originated in the region of nowadays Indochina and expanded from there in multiple waves. More details are given in *Figure 6*.

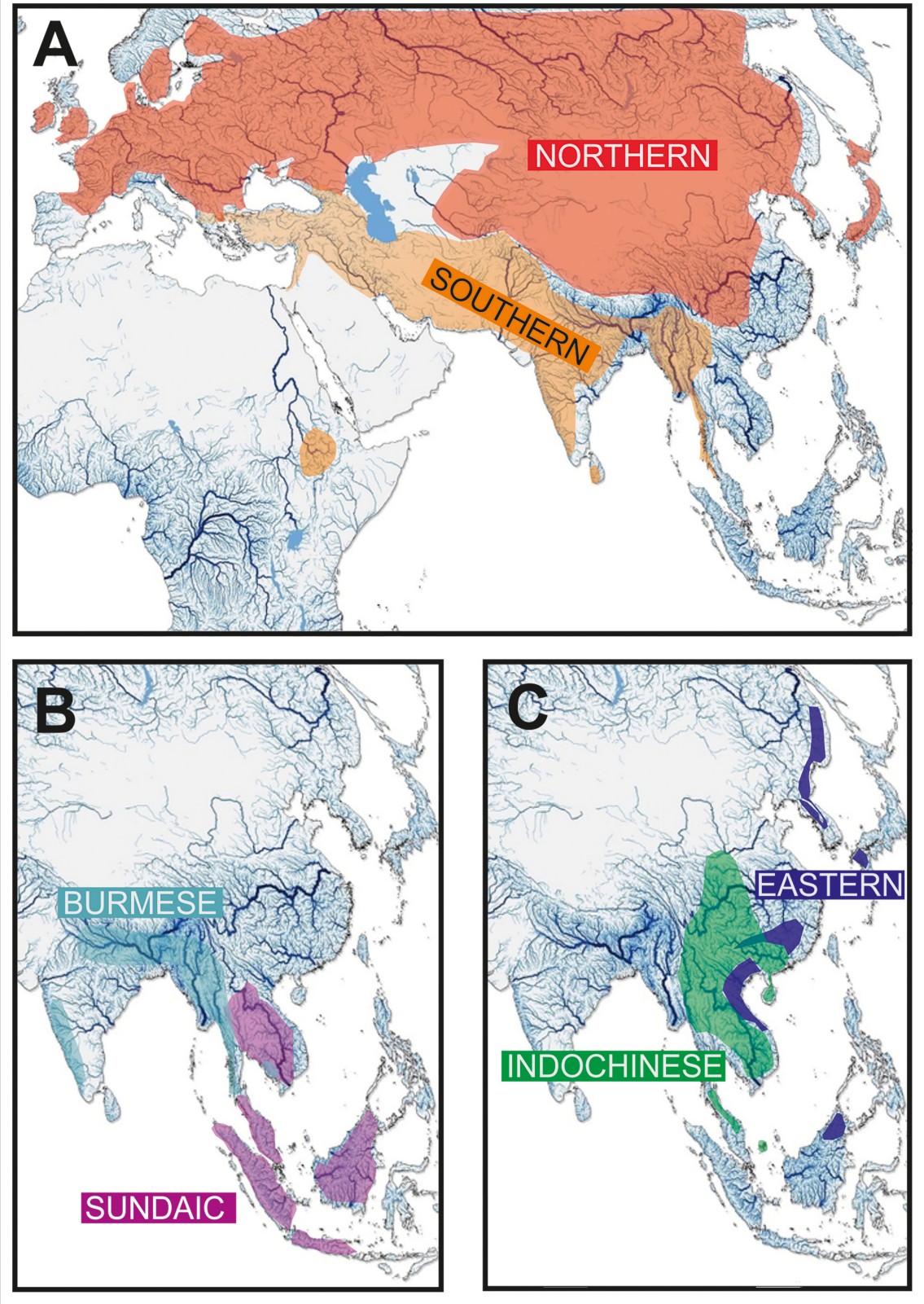

**Figure 4.** Geographic distribution of the six major clades of Nemacheilidae. (**A**) The Northernand Southern Clades, (**B**) The Burmese and Sundaic Clades, (**C**) The Eastern and Indochinese Clades.

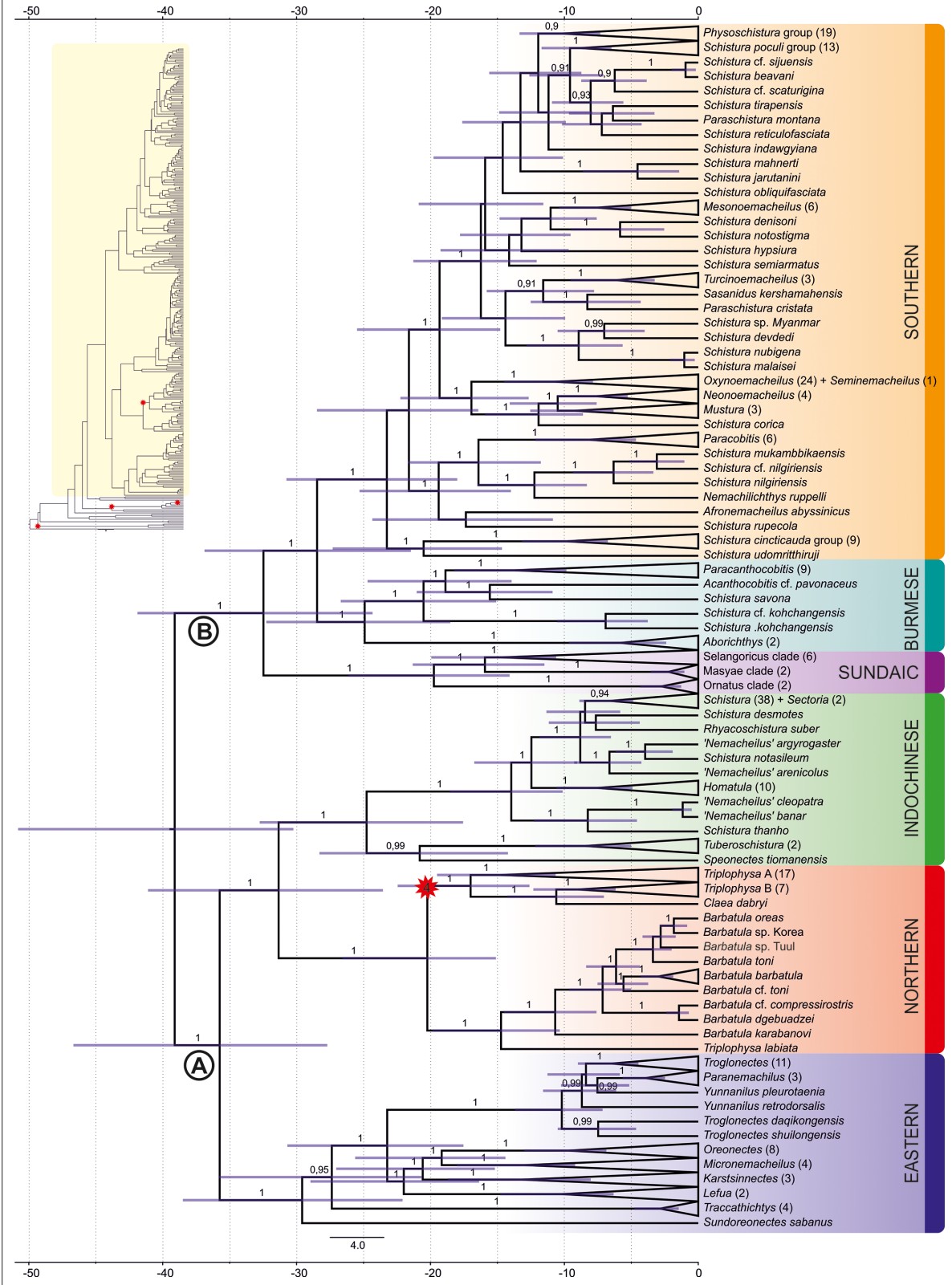

**Figure 5.** Divergence time estimation: ingroup of the maximum clade credibility (MCC) tree resulting from Bayesian divergence time analysis of concatenated dataset in BEAST 2. The red star indicates internal calibration point, the values at the nodes represent posterior probabilities (PPs lower than 0.90 not shown), and the blue bars relevant 95% HPD. The time scale is in millions of years. Inset in the top-left corner shows complete tree: Nemacheilidae is highlighted in yellow, while the outgroup is not highlighted; all calibration points are indicated by red stars.

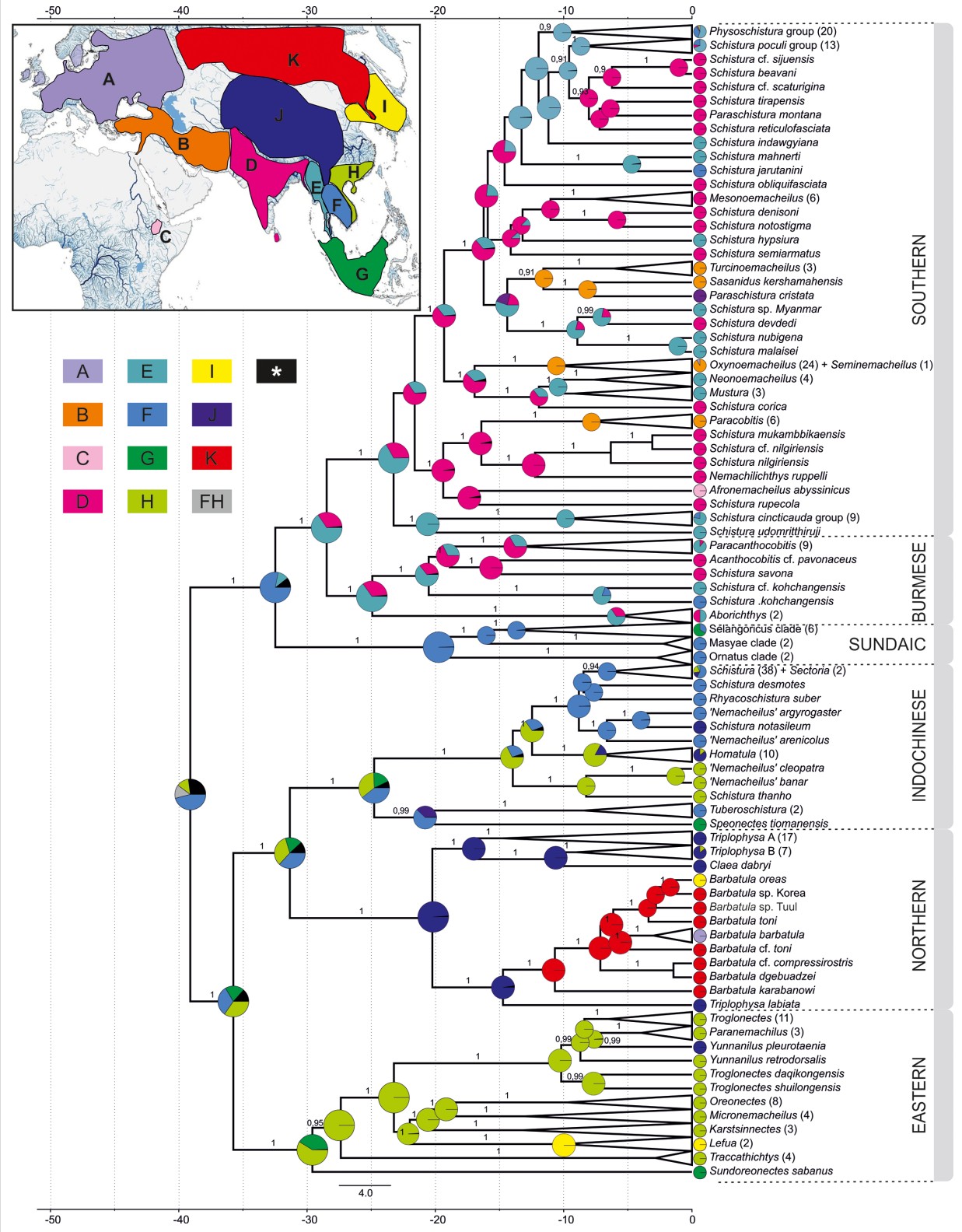

**Figure 6.** Reconstruction of the biogeographical history of Nemacheilidae. The inset map in the top-left corner shows the defined biogeographical regions of Eurasia recognised in the analysis; below the map the colour – region code for the main tree. Grey colour indicates combination of two regions as ancestral (FH); black colour indicates ancestral ranges identified with likelihood <10%. Main tree: ancestral range reconstruction with use of DEC+J model based on the phylogeny derived from BEAST2. Pie charts at the nodes indicate relative likelihood of ancestral ranges of most recent common ancestors (MRCA), while the current distribution of taxa is indicated at the tips.

## Discussion

### The evolutionary history of Nemacheilidae (Figures 7 and 8, Table 1)

#### Early Eocene (56–48 mya): The origin of Nemacheilidae (*Figure 8A*)

The family Nemacheilidae shared the last common ancestor with the other Cypriniformes about 48.5 mya (*Figure 3*), and our biogeographic analysis identified the region of nowadays Indochina as most likely place of origin of Nemacheilidae (*Figure 6*). During the early Eocene, Southeast Asia experienced a warm and perhumid climate, characterised by both highlands and lowlands with numerous rivers (*Morley, 2018*); conditions under which stone loaches flourished (*Figures 7 and 8A*; *Table 1*).

#### Middle to late Eocene (48–34 mya): First range expansions and formation of the first major clades (*Figure 8B and C*)

The first detectable cladogenetic event within the family occurred about 39 mya and separated a lineage formed by the Eastern, Northern, and Indochinese Clades (branch A in *Figure 5*) from a lineage including the Sundaic, Burmese, and Southern Clades (branch B). Our biogeographic analysis of the dataset revealed Indochina as most parsimony area of the first cladogenetic event; and consequently as region of origin for branch A as well as branch B.

The <u>Eastern Clade</u> was the first of the six present major clades to form: it separated about 35.8 mya from the common ancestor of the Indochinese and Northern Clades, either in the region of Indochina or Vietnam-south China. Taking into consideration its present-day distribution (*Figure 2*), we assume that the young Eastern Clade spread mainly along the eastern coastline of Asia southwards to Borneo and northwards to northeast China. Geological data support this hypothesis: during the Eocene, Asia was much smaller than today and had a rather flat relief with old mountains mainly along its eastern coast due to the collision with the Pacific Plate since 190 mya (*Seton et al., 2012*; *Wang, 2004*), while the upfolding of east-west mountain ridges had just started. The generally warm and humid climate during Eocene (*Morley, 2018*; *Westerhold et al., 2020*) additionally promoted the range expansion of the Eastern Clade. It can be assumed that shortly after the range expansion the distribution of the Eastern Clade was more or less continuous and not separated into several disjunct areas as nowadays.

The <u>ancestral Indochinese + Northern Clades</u> also spread northwards, but their present-day distribution (*Figure 2*) suggests that they took a route through Central Indochina and China. Most likely, this first geographic expansion led the ancestral Indochinese + Northern Clades northward to at least the Tibetan Protoplateau, which during Eocene epoch had its northern boundary in the region of today's Tanggula mountains (*Wang et al., 2008*), which means not even half of its present north-south extension. The presence of the two protoclades on the early Tibetan Plateau is indicated by the presence of some of their most basal members in the Yangtze basin and Yunnan: the genus *Homatula* for the Indochinese Clade and several species of *Triplophysa* for the Northern Clade (*Figures 1 and 5*).

#### Oligocene (34–23 mya): Range expansions, extinctions, and the formation of the remaining major clades (*Figure 8D and E*)

The Eocene–Oligocene transition brought a period of significant global cooling (*Hutchinson et al., 2021*) as well as a change in the precipitation regime of Central Asia (*Clift and Webb, 2019*; *Ye et al., 2022*) due to the ongoing uplift of the Himalayan chain (*Ding et al., 2017*). A substantial decrease of rainfall in Central Asia caused an aridification from 34 to 23 mya (*Bosboom et al., 2014*) extending across nearly all of Oligocene Asia from the western to the eastern coast (*Wang, 2004*). The distribution area of the <u>Eastern Clade</u> became fragmented during this period (*Figure 2*).

The Central Asian arid zone also split the Indochinese–Northern protoclade into two parts that evolved into two separate major clades, the <u>Indochinese Clade</u> and the <u>Northern Clade</u>. Moreover, the distribution area of the Indochinese Clade was split and separated the southernmost taxa (*Speonectes* and *Tuberoschistura*) from the taxa presently inhabiting all of Indochina and southern China, including the middle Yangtze basin (*Figure 2*). The absence of the Indochinese Clade from any waters north of the Tibetan Plateau suggests that its Oligocene refuge was located southeast of the Tibetan Protoplateau. In contrast, the early Northern Clade found its refuge further north on the Tibetan Protoplateau, the only region that can explain the further history and the present distribution of the clade. By late Oligocene, the climate in Central Asia returned to a warmer and moister state (*Zachos et al., 2001*) and the Pamir and western Kunlun Mountains (in present-day SE Tajikistan, N Pakistan,

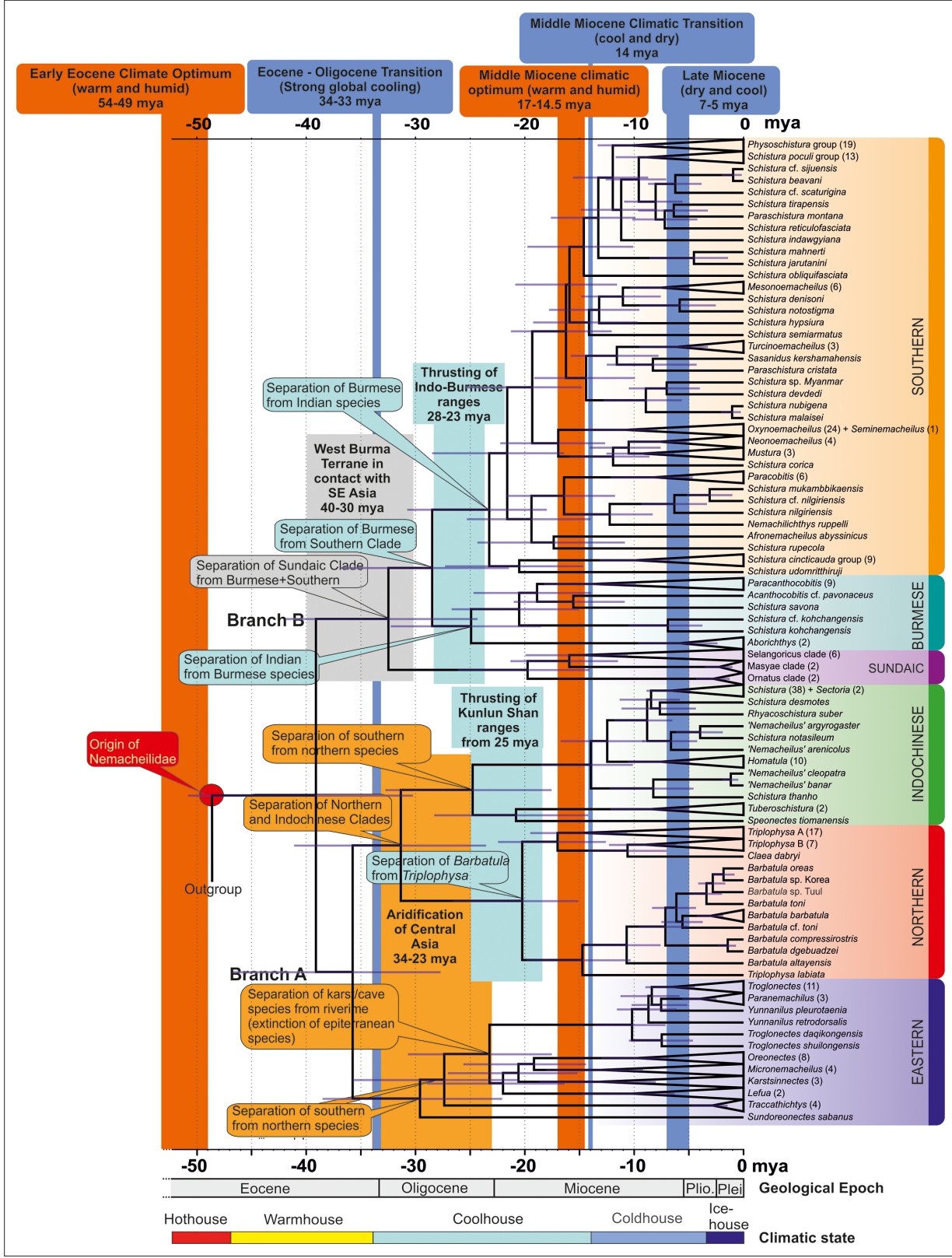

**Figure 7.** Time tree of the evolutionary history of Nemacheilidae: the figure presents the evolutionary history of Nemacheildae, highlighting the dating of key geological or climatic key paleo-events that might have significantly impacted their evolution. Orange bars represent exceptionally warm periods, blue bars denote cold periods. Boxes indicate specific geological or climatic event. For detailed explanations of these events and their impact on nemacheilid evolution, refer to text. Climatic states from *Westerhold et al., 2020*.

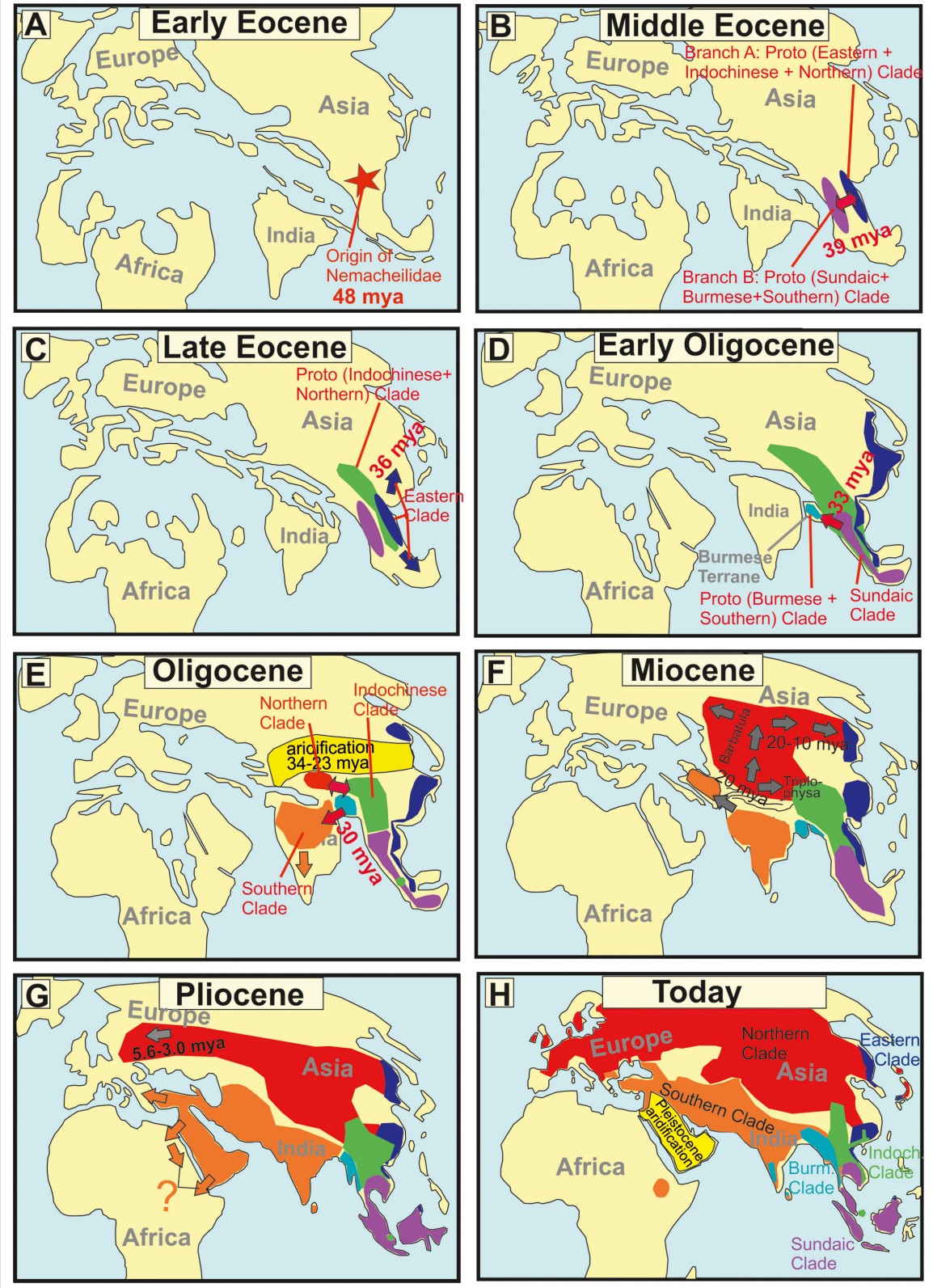

**Figure 8.** Paleomaps visualising the geographical and chronological aspects of the main events during the evolutionary history of the freshwater fish family Nemacheilidae across Eurasia. (**A**) Early Eocene, (**B**) Middle Eocene, (**C**) Late Eocene, (**F**) Miocene, (**G**) Pliocene, (**H**) Today A red star marks the point of origin, arrows indicate dispersal events. Background maps were drawn according to information given in *Blakey, 2023*, *Atlas, 2023*, *Wang, 2004*, *Westerweel et al., 2020*.

**Table 1.** Overview of the major evolutionary events in the six main clades within Nemacheilidae from its origin 48 mya until today.

| Period | Eastern | Indochinese | Northern | Sundaic | Burmese | Southern |
|---|---|---|---|---|---|---|
| Early Eocene 48 mya | Origin of Nemacheilidae in the area of present-day Indochina; probably related to a strong diversification period during the Early Eocene Climatic Optimum (54–49 mya). | | | | | |
| Middle Eocene 39 mya | Split of Nemacheilidae into two branches (Proto Eastern + Indochinese + Northern branch and Proto Sundaic +Burmese + Southern branch). The ultimate cause for this split is unknown. | | | | | |
| Late Eocene 36 mya | Formation of Eastern Clade; spreading along the eastern margin of Asia. | The Proto Indochinese + Northern Clade starts expanding northwards and southwards (at least to present-day southern Malay Peninsula). | | | | |
| Early Oligocene 33 mya | | | | Formation of the Sundaic Clade by separation of the Proto Burmese +Southern Clade. | The Proto Burmese + Southern Clade forms by colonising the Burmese Terrane, a tectonic microplate that travels northwards with the Indian plate. | |
| Oligocene 34–23 mya | The distribution of the Eastern Clade becomes fragmented by strong aridification of Central Asia. | Proto Indochinese + Northern Clade in two refuges during strong aridification of Central Asia. The Proto- Northern Clade endures the period on the northern part of the young Tibetan Plateau, the Proto-Indochinese Clade east of the Tibetan Plateau. After this long time of separation they form the Northern and Indochinese Clades. | | | The Proto Burmese +Southern Clade colonises the Indian Plate (30 mya). Burmese Terrane and India collide with Asian mainland, causing the rise of the Indo-Burmese mountain ridge (from 28 mya), separating the Southern Clade from the Burmese Clade. | |
| Early and Middle Miocene 23–10 mya | | | The genus *Barbatula* spreads northwards as the Kunlun, Pamir and Altai mountains form (from 25 mya), then eastwards to Korean Peninsula; the genus *Triplophysa* spreads eastwards into southern China. | | | The Southern Clade colonises the freshly forming Near East in several waves. The Proto-*Afronemacheilus* separates from Indian species. |
| Miocene 23–5.3 mya | Period of strong diversification, especially around the time of the Middle Miocene Climatic optimum (17–14.5 mya), Number of lineages in Nemacheilidae rises from 16 to 177. | | | | | |
| Pliocene 5–2.5 mya | | | *Barbatula* spreads westwards due to ice formation in northern Siberia and colonises Europe. In the east it colonises the Japanese Archipelago. | | | |
| Pleistocene from 2.6 mya | | | | | | Extinctions from Arab Peninsula and the lower Nile valley due to strong aridification. |

and W China) rapidly thrust upwards as an expansion of the northern Tibetan Plateau (*Zheng et al., 2015*). These favourable conditions coincided with a major split within the Northern Clade at 20 mya (*Figure 5*), when the proto-*Barbatula* clade colonised the Pamir and Kunlun Mountains, while the proto *Triplophysa* branch found its main distribution rather east of the Tibetan Plateau.

Different geological events drove the evolution of branch B: the common ancestor of the Burmese and Southern Clades split at 32.5 mya in Indochina from the <u>Sundaic Clade</u> because the former left Southeast Asia by colonising the West Burma Terrane, a microplate that was pushed northward by the Indian plate (*Morley et al., 2021*) and made temporary contact with western Sundaland (today's western Malay Peninsula) around Eocene–early Oligocene (*Westerweel et al., 2019*). Around 30 mya, it made contact with northeast India (*Morley et al., 2021*), allowing fishes from the West Burma

Terrane to colonise the Indian subcontinent. During the Oligocene, the Indian plate and the West Burma Terrane both collided with mainland Asia. The Indian plate presently forms the Indian subcontinent, the West Burma Terrane now forms western Myanmar (*Westerweel et al., 2020*). Their parallel collision with mainland Asia caused the uplift of the north–south directed Indo-Burmese mountain range (*Morley et al., 2020*), separating the aquatic fauna of the Brahmaputra basin on the Indian side, from the Chindwin-Irrawaddy basin on the Burmese side, thus separating the Burmese Clade from the Southern Clade.

## Miocene (23–5.3 mya): Rapid diversification (*Figure 8F and G*)

The Miocene is generally an epoch of cooling, with temperatures decreasing from about 5.5°C above present conditions to conditions similar to present (*Zachos et al., 2001*). But temperatures also underwent strong fluctuations, leading to exceptionally warm periods like the Middle Miocene Climatic Optimum (17–14.5 mya), as well as to drastic cooling events like the Middle Miocene Climatic transition (14 mya) (*Morley, 2018*). Additionally, the Asian monsoon regime strengthened from the early Miocene (*Clift and Webb, 2019*), increasing precipitation and altering the distribution of climatic regions across Asia (*Sun and Wang, 2005*). Moreover, during the Miocene the Himalayan Mountains experienced their main uplift (*Ding et al., 2017*) and the upfolding area of Central Asia mountain ranges expanded northwards, giving rise to the Tian Shan and Altai ranges (*Nissen et al., 2009*; *Zheng et al., 2015*; *Liu et al., 2023*).

During this complex geological and climatic scenario, the family Nemacheilidae underwent a massive diversification. In our phylogenetic hypothesis, the family comprised 16 lineages at the beginning of Miocene. These lineages diversified during Miocene into 177 lineages. The radiation occurred across all six major clades, albeit to a lesser extent in the Eastern Clade. Interestingly, it manifested similarly in the region of the Indian monsoon and the East Asian monsoon. In both regions, major radiations occurred during the Middle Miocene Climatic Optimum, including the radiation of the *Barbatula* lineage in the Northern Clade, of the Indochinese Clade, the Sundaic Clade and of six branches in the Southern Clade. It appears that under increased temperature and precipitation freshwater fishes flourished, across all of Asia and under different monsoon systems.

One of the radiations that began during the Middle Miocene Climate Optimum involves *Afronemacheilus*, the sole nemacheilid genus found in Africa (*Prokofiev and Golubtsov, 2013*). Our results confirm that the ancestral *Afronemacheilus* separated from the Southern Clade about 18–17 mya. It could have utilised two potential pathways to Africa: one across the Arabian Peninsula and via the southern Danakil Isthmus, a broad landbridge that stretched from Djibouti to Aden from late Miocene to Pliocene (*Bosworth et al., 2005*; *Stewart and Murray, 2017*). The second pathway would lead through the Levant and via Sinai into the Nile ('*Gomphotherium* landbridge'; *Harzhauser et al., 2007*), and subsequently upstream into the Ethiopian highlands. Unfortunately, the nearly complete extinction of freshwater fishes from the Arabian Peninsula and from the Nile during Pleistocene aridification (*Kotwicki and Al Sulaimani, 2009*; *Leplongeon, 2021*) complicates the reconstruction of its pathway to Africa.

In the Northern Clade, the *Triplophysa* lineage flourished during the Miocene. Its present wide distribution in southern and Central China suggests that it expanded from its Oligocene refuge on the eastern part of the Tibetan Plateau in an eastern direction (*Figure 2*). In contrast, the *Barbatula* branch expanded from its refuge on the western Tibetan Plateau northwards, traversing the uplifting Tian Shan and Altai ranges. This expansion can be traced from the geographic position of the most basal species (*T. labiata* – Tian Shan, *B. altayensis* – Altai), and then eastwards (*B. compressirostris* + *B. dgebuadzei* – west Mongolia, *B.* cf. *toni* – east Mongolia; tip species in Korea [*B.* indet Korea] and Japan [*B. oreas*]).

During late Miocene, *Barbatula* also expanded westwards through Siberia and colonised Europe between 5.6 mya (separation of the European taxa from the Mongolian sister taxon) and 3.0 mya (oldest split within European *Barbatula*). During this cool and dry period, glaciations occurred on the northern hemisphere (*Zachos et al., 2001*), and in Siberia the mean temperature of the coldest month dropped below 0° for the first time during Cenozoic (*Popova et al., 2012*). Freezing of the northern margin of Eurasia prevented the Siberian rivers (which flow from south to north) from discharging their water into the Northern Ocean and produced extensive wetlands on the shallow relief of Siberia. These wetlands served as pathways for freshwater fishes from East Asia to European river systems

during Pleistocene (*Kalous et al., 2012*), and we assume that the same mechanism brought *Barbatula* to Europe during Pliocene.

## Pliocene and Pleistocene (5.3–0.01 mya): Ice ages

During Pliocene and Pleistocene, global cooling continued until massive glaciations repeatedly covered the north of Eurasia (*De Schepper et al., 2014*). However, the Pliocene also brought a much stronger seasonality to Eurasia, especially in the form of cold winters (*Shen et al., 2018*). Despite these significant changes in climatic conditions, the evolution of Nemacheilidae seems not to have experienced any dramatic negative impact, but reveals mainly local speciation events (e.g. the genus *Barbatula* colonised Eastern Siberia, Japan, and the Korean Peninsula and the European *Barbatula* diversified geographically). We know from more detailed studies on Nemacheilidae (*Bohlen et al., 2020a*; *Bohlen et al., 2020b*; *Šlechtová et al., 2021*) that during Pliocene and Pleistocene also geographic shifts as well as expansions and disappearances occurred, but since these happened on rather local scale they were too insignificant to reflect in the present intercontinental dataset.

## Taxonomic implications

Besides its conclusions on biogeography and evolutionary history of the Eurasian freshwater fauna, the present dataset offers insights into the taxonomy of Nemacheilidae. Among the 37 analysed genera, 11 are represented by a single species in the dataset; 18 of the genera with several species in the dataset appear as monophyletic units in our analysis. However, seven genera (*Mustura, Paraschistura, Physoschistura, Oxynoemacheilus, Schistura, Sectoria, Triplophysa*) are identified as para- or polyphyletic based on our results. The most scattered genus is *Schistura*, which is listed in three of the six major clades and in a total of about 20 different genetic lineages.

In fact, this is nothing new. The 'polyphyly' of several larger genera, particularly *Schistura*, has been recognised for a long time: already *Kottelat, 1990* referred to *Schistura* and *Nemacheilus* as 'wastebasket names'. These are not really cases of polyphyly; it is simply that a large number of new species have been described in the last 40 years and, not knowing in which genus to place them, authors simply 'dumped' them in a few catch-all genera, waiting for more comprehensive studies and knowing well that the used genus names are mere labels. In recent years, more detailed morphological studies have resolved the relations of some groups of species within these catch-all genera (e.g. *Conway and Kottelat, 2023*; *Dvořák et al., 2023*; *Kottelat, 2018*; *Kottelat, 2019*). Moreover, every genetic study that included more than five species of *Schistura* resulted in polyphyly of the genus (e.g. *Chen et al., 2019*; *Min et al., 2023*; *Sember et al., 2015*; *Sgouros et al., 2019*; *Tang et al., 2006*). A recent detailed phylogenetic investigation of the genus *Nemacheilus Šlechtová et al., 2021* found the genus to be polyphyletic, but it became monophyletic after identifying five species that were falsely placed into *Nemacheilus* (these species are labelled '*Nemacheilus*' in the present study). Similarly, revisionary studies of the genera *Paracanthocobitis, Troglonectes, Micronemacheilus, Paranemachilus,* and *Oreonectes* also allowed to resolve their intrarelationships and redefine them into monophyletic taxa (*Singer and Page, 2015*; *Luo et al., 2023*). The main obstacles to a comprehensive taxonomic revision of Nemacheilidae is its vast size, a fast increasing number of new species, and the difficulty to access material from critical areas because of international and local political issues. We believe that results of the present study will facilitate future research on the taxonomic understanding of Nemacheilidae.

Another aspect of Nemacheilidae taxonomy is the existence of numerous undescribed species. Among the 279 species in our dataset, 36 (13%) were either clearly undescribed ('*Genus* sp.') or named with reservations ('*Genus*' cf. *xxx*'). This ratio is an underestimate, since, in general, undescribed species were avoided when composing the dataset. However, the existence of a vast amount of undescribed species within Nemacheilidae is illustrated by the fact that 430 of the 838 valid species of Nemacheilidae known in November 2024 (51%) have been described in the last 25 years (pers. obs.; *Kottelat, 2012*; *Kottelat, 2013*, updated).

## Conclusions

- We reconstruct the evolutionary history of the Eurasian freshwater fish family Nemacheilidae during most of Cenozoic era. The extensive dataset (471 specimens, 279 species, 6 genes) from

across Eurasia and the low dispersal capacity of the model resulted in high resolution of the phylogenetical and biogeographical analysis.

- Molecular phylogeny uncovered six major clades within the family. Dating of cladogenetic events and ancestral range estimation traced the origin of Nemacheilidae to Indochina around 48 mya.
- Major expansions were facilitated by tectonic connections, favourable climatic conditions, and orogenic processes. Conversely, aridification emerged as the primary cause of extinction events.
- Our study represents the first comprehensive and holistic reconstruction of the evolution of Eurasian freshwater biodiversity on a continental scale and through deep geological time.
- Our analyses identified numerous polyphyletic or paraphyletic taxa, highlighting the need for a thorough taxonomic revision within the family Nemacheilidae.

## Methods

**Key resources table**

| Reagent type (species) or resource | Designation | Source or reference | Identifiers | Additional information |
|---|---|---|---|---|
| Biological sample (Nemacheilidae) | DNA sequences | GenBank | | 104 different species from the fish family Nemacheilidae (details in *Supplementary file 1*) |
| Biological sample (Nemacheilidae) | Tissue samples | This paper | | 175 different species from the fish family Nemacheilidae (details in *Supplementary file 1*) |
| Sequence-based reagent | PCR primers | | | See *Supplementary file 2* |
| Commercial assay or kit | DNeasy Blood and Tissue kit | QIAGEN | | DNA isolation kit |
| Software, algorithm | DNA Star | LASERGENE | | |
| Software, algorithm | BioEdit | *Hall, 1999* | | |
| Software, algorithm | PhyloSuite v1.2.2 | *Zhang et al., 2020* | | |
| Software, algorithm | MrBayes 3.2.7a | *Ronquist and Huelsenbeck, 2003* | | |
| Software, algorithm | ASTRAL III | *Zhang et al., 2018* | | |
| Software, algorithm | BEAST 2.6.4 | *Bouckaert et al., 2014* | | |

### Sampling and laboratory procedures

Altogether, 492 samples were examined, including 21 outgroup species and 471 nemacheilid samples covering 279 species from 37 genera of the family Nemacheilidae, including the set of sequences of 364 specimens analysed in our laboratory and sequences of 107 specimens obtained from GenBank. Tissue samples used for the present study were fixed and stored in 96% ethanol. For more details about species, geographical origin, and GenBank accession numbers, see *Supplementary file 1*.

Total genomic DNA was isolated using the DNeasy Blood and Tissue kit (QIAGEN) following the manufacturer's instructions. One mitochondrial gene (cytochrome *b*) and five nuclear genes (RAG1, IRBP2, MYH6, RH1, and EGR3) were selected. For the list of primers, see *Supplementary file 2*, and details about the PCR settings are described in *Chen et al., 2003*; *Chen et al., 2008* and *Dvořák et al., 2022*.

Sequencing of the PCR products was performed at the Laboratory of Fish Genetics on ABI Prism 3130 GA or via sequencing service in Macrogen Europe BV (Amsterdam, Netherlands).

### Phylogenetic analyses

Chromatograms were checked and assembled in the SeqMan II module of the DNA Star software package (LASERGENE). Single-gene alignments were done in BioEdit (*Hall, 1999*) with use of ClustalW (*Larkin et al., 2007*) multiple alignment algorithm. The datasets were concatenated in PhyloSuite v1.2.2 (*Zhang et al., 2020*).

The best-fit substitution models and partitioning schemes (*Supplementary file 3*) were estimated using Partition Finder 2 (*Lanfear et al., 2016*) implemented in PhyloSuite 1.2.2 based on the corrected Akaike information criterion (AICc).

The 492 specimen phylogenetic trees were inferred from six loci concatenated dataset using the ML and BI. The ML analyses were performed using IQ-TREE (*Nguyen et al., 2015*) implemented in PhyloSuite. The best-fit evolutionary model for each codon partition was automatically determined with ModelFinder (*Kalyaanamoorthy et al., 2017*). The node support values were obtained with 5000 ultrafast bootstrap replicates (UFBoot) (*Hoang et al., 2018*). For the BI analyses, we used MrBayes 3.2.7a (*Ronquist and Huelsenbeck, 2003*) on CIPRES Science Gateway (*Miller et al., 2010*). The datasets were partitioned into genes and codon positions and analyses were performed in two parallel runs of 10–20 million generations with eight Metropolis Coupled Markov Chains Monte Carlo. The relative burnin of 25% was used, and from the remaining trees a 50% majority rule consensus trees was constructed.

For reconstructing of the species tree, we used the Accurate Species TRee ALgorithm (ASTRAL III) (*Zhang et al., 2018*). For ASTRAL, we used unrooted single gene ML trees reconstructed in IQ-TREE implemented in PhyloSuite. We have used IQ-TREE 2 (*Minh et al., 2020*) to calculate gene concordance factor (gCF, indicating the percentage of decisive gene trees containing particular branch) and site concordance factor (sCF, indicating the percentage of decisive alignment sites that support a branch in the reference tree) for every branch in the ASTRAL species tree. Distribution maps in *Figure 3* were designed using HYDROSHEDS (*Lehner and Grill, 2013*). All original trees are available from authors upon request.

## Divergence time estimations and ancestral range reconstruction

The ages of cladogenetic events were estimated in BEAST 2.6.4 (*Bouckaert et al., 2014*) via CIPRES Science Gateway with use of four calibration points. The first calibration point is based on the oldest known fossil of the family Catostomidae, *Wilsonium brevipinne*, from early Eocene (*Liu, 2021*), approximately 56–48 my old. The second calibration point is derived from the only known nemacheilid fossil record, *Triplophysa opinata* from Kyrgyzstan from middle-upper Miocene (16.0–5.3 mya) (*Böhme and Ilg, 2003*; *Prokofiev, 2007*). For some time, the Miocene fossil species *Nemachilus tener* from Central European was considered a member of Nemacheilidae. However, subsequent research has cast doubt on placement of this species within loaches (*Obrhelova, 1967*), leaving *T. opinata* as the only known fossil of Nemacheilidae. Third calibration point is based on the oldest known fossil of the genus *Cobitis*, *C. naningensis*, from early to middle Oligocene in southern China, 34–28 my old (*Chen et al., 2015*). The fourth calibration point, based on the river history of the southern Korean Peninsula, dates the separation between the species *Cobitis tetralineata* and *C. lutheri* to 2.5–3.5 my (*Kwan et al., 2014*). For the fossil-based calibration points, we have used a fossil calibration prior implemented in CladeAge (*Matschiner et al., 2017*), the fourth calibration point was set to a uniform prior (2.5–3.5). The calibration points are depicted in *Figures 3 and 5*.

For the final analyses, the partitions were unlinked and assigned the estimated evolutionary models. We used relaxed lognormal molecular clock and birth-death prior. Given the large and complex nature of our dataset, and in light of several preliminary analyses where MCMC chains did not provide satisfactory effective sampling sizes (ESSs) even after 1 billion iterations, we made the decision to streamline the dataset by retaining only one specimen per species. To enhance the performance of the analysis, we also implemented parallel tempering using the CoupledMCMC package (*Müller and Bouckaert, 2020*; *Müller and Bouckaert, 2019*). The analysis was configured with four parallel chains of $6 \times 10^8$ generations, with resampling every 1000. Tree and parameter sampling intervals were set to 60.000. The ESSs for all parameters were assessed in Tracer 1.7.1 (*Rambaut et al., 2018*). A maximum clade credibility (MCC) tree was built in TreeAnnotator 2.6.0 (*Rambaut and Drummond, 2010*) after discarding the first 25% of trees. The final trees were visualised in FigTree 1.4.4 (*Rambaut, 2019*). ML and BI were performed with both full as well as the reduced datasets.

The biogeographical analysis was conducted using the BioGeoBEARS package (*Matzke, 2013*) implemented in RASP 4.0 (Reconstruct Ancestral Stage in Phylogenies, *Yu et al., 2015*). We set 11 biogeographic regions according to drainage areas separated by major mountain ridges (Ural, Caucasus. Himalayas, Tenasserim Ridge, Indo-Burmese Range, Annamite Cordilliera, Balkan, Hindukush) or saltwater straits (Red Sea) (see *Figure 4A*). The selected biogeographic regions are mostly congruent with the classical subregions of the Palaearctic (European, Siberian, East Asia, and Mediterranean) and the Oriental (Indian, Malay, and Chinese). For the analysis, we utilised the final MCC tree and 100 post burnin trees from the previous BEAST analysis. A maximum of two-unit areas

was allowed to be present in every reconstructed ancestral range. The comparison of six biogeographic models (DEC, DEC+J, DIVALIKE, DIVALIKE+J, BAYAREALIKE, BAYAREALIKE+J) suggested that the DEC+J is the best-suited model for the given dataset (as indicated by AICc weight value, *Supplementary file 4*). Likelihood ratio test (LRT) p-values, comparing models with and without the J factor, indicated that the addition of the J factor significantly influences the model likelihood. Considering recent criticisms of DEC and DEC+J (*Ree and Sanmartin, 2018*), we opted to conduct the analysis not only with DEC+J but also with other +J models. Comparison of the results from all applied models revealed only minimal differences.

## Acknowledgements

VŠ, TD, VŠ and JB would like to thank R Hoyer, J Kühne, FF Kullander (NRM), J Kuszniersz, K Lim (ZRC), H Linke, A Nolte, G Ott, M Reichard, I Seidel, S Somadee, HH Tan (ZRC), TK Toe, K Udomritthiruji, T Win, and U Wolf for their help with obtaining specimens for this study and Š Pelikanová for help with laboratory work. VŠ and JB were funded by grants 206/08/0637 and 19-18453S of the Czech Science Foundation. BL and AG greatly acknowledge S Cherenkov for his help in sampling. BL and AG was supported by the Russian Science Foundation, grant no. 24-44-20019. Materials from Laos, Myanmar, and Mongolia were obtained by MK under various contracts and over the years he was assisted mainly by Nyein Chan, T Phommavong, and the late M Erdenebat; the late T Whitten has been essential in creating these great opportunities.

## Additional information

### Funding

| Funder | Grant reference number | Author |
|---|---|---|
| Czech Science Foundation | 19-18453S | Vendula Bohlen Šlechtová Joerg Bohlen |
| Russian Science Foundation | 24-44-20019 | Boris Levin Alexander Golubtsov |
| Czech Science Foundation | 206/08/0637 | Vendula Bohlen Šlechtová Joerg Bohlen |

The funders had no role in study design, data collection and interpretation, or the decision to submit the work for publication.

### Author contributions

Vendula Bohlen Šlechtová, Conceptualization, Formal analysis, Supervision, Visualization, Methodology, Writing – review and editing; Tomáš Dvořák, Data curation, Formal analysis, Investigation; Jörg Freyhof, Maurice Kottelat, Boris Levin, Resources, Writing – review and editing; Alexander Golubtsov, Resources; Vlastimil Šlechta, Data curation, Formal analysis, Writing – review and editing; Joerg Bohlen, Conceptualization, Resources, Formal analysis, Visualization, Methodology

### Author ORCIDs

Vendula Bohlen Šlechtová ⓘ https://orcid.org/0000-0002-5499-7550
Boris Levin ⓘ https://orcid.org/0000-0002-4044-2036
Joerg Bohlen ⓘ https://orcid.org/0000-0003-1259-008X

Reviewer #2 (Public review): https://doi.org/10.7554/eLife.101080.3.sa1
Author response https://doi.org/10.7554/eLife.101080.3.sa2

## Additional files

### Supplementary files

Supplementary file 1. List of analysed samples, their identification, geographical origin, voucher

number, and GenBank accession numbers for their sequences. Voucher numbers starting with 'A' refer to the collection of IAPG, Liběchov, Czech Republic; 'CMK' numbers refer to the collection of Maurice Kottelat; 'GenBank' refers to sequences from GenBank; 'ZRC' to samples housed in the Lee Kong Chian Natural History Museum, National University of Singapore, Singapore.

Supplementary file 2. List of primers used in the present study for amplification and/or sequencing.

Supplementary file 3. Alignment attributes and best-fit models. Lengths of alignments, numbers of variable (VP), and parsimony informative (PI) positions and models estimated for all partitions. BEAST and MrBayes models were calculated in Partition Finder 2 (PF2, *Lanfear et al., 2016*) implemented in PhyloSuite v1.2.2 (*Zhang et al., 2020*) under AICc criterion, with greedy algorithm (*Lanfear et al., 2016*) and branch lengths linked. For ML trees, the models and partitioning schemes were estimated under BIC with ModelFinder (*Kalyaanamoorthy et al., 2017*) implemented in IQ tree. The values and models were calculated for both (A) full as well as (B) reduced dataset. Table (C) provides an overview of data attributes for the ingroup dataset only.

Supplementary file 4. Comparison of biogeographic models in RASP. The last column shows the p-values of the likelihood ratio test. Based on AICc weight (AICc wt), the DEC+J model is recommended as the best fit for our dataset, supported by low p-values indicating the significant influence of the J factor on model likelihood.

MDAR checklist

## Data availability

Original sequences have been deposited in GenBank under the accession numbers PP259624–PP259994, PP279842–PP280528 and PP315675–PP315895. Details about data are given in *Supplementary files 1 and 2*. All original trees are deposited to Dryad.

The following dataset was generated:

| Author(s) | Year | Dataset title | Dataset URL | Database and Identifier |
|---|---|---|---|---|
| Šlechtová VB, Dvorak T, Freyhof J, Kottelat M, Levin B, Golubtsov A, Slechta V, Bohlen J | 2025 | Reconstructing the phylogeny and evolutionary history of freshwater fishes (Nemacheilidae) across Eurasia since early Eocene | https://doi.org/10.5061/dryad.rxwdbrvhz | Dryad Digital Repository, 10.5061/dryad.rxwdbrvhz |

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
