## [Editor Report · eLife Assessment]

In this **important** study, the authors reconstruct the evolutionary history of a large and widespread group of freshwater fishes (Nemacheilidae) across Eurasia since the early Eocene, based on molecular phylogenetic analysis with very comprehensive samplings including 471 specimens belonging to 250 living species. The authors **convincingly** infer that range expansions of the family were facilitated by tectonic connections, favourable climatic conditions, and orogenic processes, adding to our understanding of the effects of climatic change on biodiversity during the Cenozoic. This work is of interest to evolutionary biologists, ichthyologists, paleontologists, and general readers.

---

## [Referee Report · Reviewer #2 (Public review)]

Summary:

The authors present the results of molecular phylogenetic analysis with very comprehensive samplings including 471 specimens belonging to 250 species, trying to give a holistic reconstruction of the evolutionary history of freshwater fishes (Nemacheilidae) across Eurasia since the early Eocene.

Strengths:

They provide very vast data and conduct comprehensive analysis. They suggested that Nemacheilidae contain 6 major clades, and the earliest differentiation can be dated to early Eocene.

Weaknesses:

They did not discuss the systematic problems widely existing, did not use the conventional way to discuss the evolutionary process of branches or clades, but just chronically describe the overall history.

Comments on revisions:

As the authors are aware that there are some taxonomic problems, which can not be solved at present. And they have mentioned this in the revised manuscript. I can not provide other suggestions at the moment.

---

## [Author Response]

The following is the authors’ response to the original reviews.

**Reviewer #1 (Public review):**
Summary:This is by far the phylogenetic analysis with the most comprehensive coverage for the Nemacheilidae family in Cobitoidea. It is a much-lauded effort. The conclusions derived using phylogenetic tools coincide with geological events, though not without difficulties (Africa pathway).Strengths:Comprehensive use of genetic toolsWeaknesses:Lack of more fossil records

Thank you for appreciating the comprehensiveness of our study.

We agree that additional nemacheilid fossils would have provided valuable support for reconstructing the evolutionary history of the family. However, the nemacheilid fossil used in our study is currently the only fossil species of the family, which precludes the possibility of including more. To address this limitation, we incorporated fossils from closely related fish families, as well as a geological event, to calibrate the time tree. We have added further details on this point in “Divergence time estimations and ancestral range reconstruction” section of the Methods. The reconstruction of the pathway by which loaches reached northeast Africa, is further complicated by the extensive aridification of the Arabian Peninsula and the Nile valley, leaving no fossil or extant Nemacheilidae species of Nemacheilidae to provide insights into the distribution of the family during late Miocene.

**Reviewer #2 (Public review):**
Summary:The authors present the results of molecular phylogenetic analysis with very comprehensive samplings including 471 specimens belonging to 250 species, trying to give a holistic reconstruction of the evolutionary history of freshwater fishes (Nemacheilidae) across Eurasia since the early Eocene. This is of great interest to general readers.Strengths:They provide very vast data and conduct comprehensive analyses. They suggested that Nemacheilidae contain 6 major clades, and the earliest differentiation can be dated to the early Eocene.Weaknesses:The analysis is incomplete, and the manuscript discussion is not well organized. The authors did not discuss the systematic problems that widely exist. They also did not use the conventional way to discuss the evolutionary process of branches or clades, but just chronologically described the overall history.

In the revised version, we address the systematic issues within Nemacheilidae in a new paragraph. The polyphyly of the genus Schistura and the polyphyly or paraphyly of many other nemacheilid genera are wellknown challenges in ichthyology. However, the large size of the family Nemacheilidae and the absence of a clear basal classification system has made systematic work difficult.

The chronological concept in the description of events is in accordance with the sequence in which the events occurred over time and corresponds with Figure 8. Additionally, a clade-by-clade description would make it challenging to capture the periods before all clades were formed. As a compromise, the revised version includes a new table where each clade is represented by a column, allowing readers to trace the history of each clade in a clear overview. With this table, we make both the chronological and clade-by-clade perspectives to enhance reader understanding

**Recommendations for the authors:**

**Reviewer #1 (Recommendations for the authors):**
I have no major comments, except for Figure 8, where the colour code for Sunda is not consistent, appearing as light purple and then dark purple. I was trying to locate the colour legend, maybe include this for all figures or refer to it.

Figure 8 has been revised to improve matching of the colours.

**Reviewer #2 (Recommendations for the authors):**
(1) It is better to discuss the evolutionary history of the major inner groups. For example, why the Branch A and B differentiated? How are the 6 major clades differentiated?

As mentioned above, the new table provides an overview of the evolutionary history of the major clades and, where known, the mechanism that led to their differentiation. For branches A and B, the underlying causes of differentiation remain known. Currently, the extensive morphological variability within each clade prevents a definitive morphological diagnosis, but such a study is planned for the future.

(2) In this study, there are still some phylogenetic or systematic problems unresolved. For example, the Genus Schistura remains polyphyletic even in different major clades. The situation is similar for the Genus Tripophysa though not so serious. These need to be discussed or at least partially solved before discussing the evolutionary history.

We discuss these topics now in a new paragraph ‘Taxonomic implications’.

(3) In Table S1, what is the meaning of "-". Does this mean no data available? If so, how do the authors treat this in their phylogenetic analysis?

Indeed, in Table S1, a ‘-‘ indicates that no sequence was available for the given species and gene. In the phylogenetic analyses, these cases were treated as missing data.

(4) What is the source of Figure 8? There are different opinions on the geological events. The authors need to indicate the source of their information.

The sources of Fig. 8 are now provided in the figure caption.

(5) The Eastern Clade forms continuous distribution in Figure 6, but discontinuous in Figure 8. Is this correct?

Figure 6 does not display the distribution areas for the clades, but illustrates the biogeographic regions used in the biogeographic analysis.